# Suppression of IFN-γ Production in Murine Splenocytes by Histamine Receptor Antagonists

**DOI:** 10.3390/ijms19124083

**Published:** 2018-12-17

**Authors:** Miho Kamei, Yukie Otani, Hidenori Hayashi, Tadaho Nakamura, Kazuhiko Yanai, Kazuyuki Furuta, Satoshi Tanaka

**Affiliations:** 1Department of Immunobiology, Okayama University Graduate School of Medicine, Dentistry, and Pharmaceutical Sciences, Tsushima naka 1-1-1, Kita-ku, Okayama 700-8530, Japan; pqfa8kfp@s.okayama-u.ac.jp (M.K.); ph20107@s.okayama-u.ac.jp (Y.O.); ph19130@s.okayama-u.ac.jp (H.H.); furutak@okayama-u.ac.jp (K.F.); 2Division of Pharmacology, Faculty of Medicine, Tohoku Medical and Pharmaceutical University School of Medicine, 1-15-1 Fukumuro, Miyagino-ku, Sendai 983-8536, Japan; tadahonakamura@tohoku-mpu.ac.jp; 3Department of Pharmacology, Tohoku University Graduate School of Medicine, 2-1 Seiryo-machi, Aoba-ku, Sendai 980-8575, Japan; yanai@med.tohoku.ac.jp; 4Department of Pharmacology, Kyoto Pharmaceutical University, Misasagi Nakauchi-cho 5, Yamashina-ku, Kyoto 607-8414, Japan

**Keywords:** IFN-γ, histamine, splenocyte, histamine H_1_ receptor, histidine decarboxylase

## Abstract

Accumulating evidence suggests that histamine synthesis induced in several types of tumor tissues modulates tumor immunity. We found that a transient histamine synthesis was induced in CD11b^+^Gr-1^+^ splenocytes derived from BALB/c mice transplanted with a syngeneic colon carcinoma, CT-26, when they were co-cultured with CT-26 cells. Significant levels of IFN-γ were produced under this co-culture condition. We explored the modulatory roles of histamine on IFN-γ production and found that several histamine receptor antagonists, such as pyrilamine, diphenhydramine, JNJ7777120, and thioperamide, could significantly suppress IFN-γ production. However, suppression of IFN-γ production by these antagonists was also found when splenocytes were derived from the *Hdc*^−/−^ BALB/c mice. Suppressive effects of these antagonists were found on IFN-γ production induced by concanavalin A or the combination of an anti-CD3 antibody and an anti-CD28 antibody in a histamine-independent manner. Murine splenocytes were found to express H_1_ and H_2_ receptors, but not H_3_ and H_4_ receptors. IFN-γ production in the *Hh1r*^−/−^ splenocytes induced by the combination of an anti-CD3 antibody and an anti-CD28 antibody was significantly suppressed by these antagonists. These findings suggest that pyrilamine, diphenhydramine, JNJ7777120, and thioperamide can suppress IFN-γ production in activated splenocytes in a histamine-independent manner.

## 1. Introduction

Histamine has diverse physiological roles, which are mediated by four subtypes of its specific receptors [1]. They are excellent and promising targets of therapeutic compounds for various diseases, such as type I allergies, peptic ulcers, and sleep disorders. Histamine is formed through the decarboxylation of histidine by l-histidine decarboxylase (HDC), which is expressed in limited kinds of cells, such as mast cells, enterochromaffin-like cells, and histaminergic neurons. Accumulating evidence suggests that HDC could be induced in some kinds of myeloid cells in addition to mast cells [2,3,4,5]. Activated neutrophils have also been identified as the source of histamine, which might be involved in lung and airway inflammation in response to mycoplasma pneumonia infection [6,7].

The question of how histamine # modulates tumor immunity has been controversial. Jutel et al. demonstrated, using *Hh2r*^−/−^ mice, that H_2_ receptors expressed in T cells are involved in the suppression of their cytokine production [8]. We previously raised the possibility that histamine may suppress tumor immunity through inhibition of the local expression of several cytokines that have the potential to antagonize tumor growth by acting on the H_2_ receptors in a murine syngeneic tumor model [9,10]. Granulocytic myeloid cells expressing HDC were found to recruit FoxP3^+^ cells in murine colon cancer [11]. These findings suggest that histamine contributes to the suppression of immune responses. On the other hand, Yang et al. demonstrated that the frequency of chemical carcinogenesis was increased in *Hdc*^−/−^ mice [12]. They hypothesized that histamine synthesis induces maturation of CD11b^+^Ly6G^+^ cells in an autocrine manner, resulting in impaired immunosuppressive function. This concept has also been supported by the findings of Ahn et al. in a murine glioma model [13]. They suggested that suppression of the activity of the myeloid-derived suppressor cells (MDSC) by histamine might result in enhanced tumor immunity. However, the mechanism of histamine-mediated enhancement of tumor immunity remains to be clarified. Further investigations should be performed to determine how HDC is induced and which subtypes of histamine receptors are involved during MDSC maturation.

It is of great significance to clarify how histamine modulates tumor immunity, because various histamine receptor ligands have been developed and some of them are approved for clinical use. We found that histamine synthesis was induced in murine splenocytes derived from tumor-bearing mice when they were co-cultured with tumor cells. We here determined the identity of histamine-forming cells in the splenocytes and investigated the roles of histamine using various histamine receptor antagonists.

## 2. Results

### 2.1. Induced Histamine Synthesis in Splenocytes Derived from Tumor-Bearing Mice

We previously demonstrated that histamine synthesis was induced in the tumor tissue in a murine syngeneic tumor model, in which CT-26 cells were transplanted in the dorsal skin of male BALB/c mice [9]. We then asked whether splenocytes derived from tumor-bearing mice could produce histamine in the presence of CT-26 cells in vitro to identify the population responsible for histamine synthesis in tumor immunity. Significant levels of HDC activity were observed in the splenocytes derived from the tumor-bearing mice when they were co-cultured with mitomycin c-treated CT-26 cells (Figure 1a). Histamine synthesis was detected neither in the splenocytes alone nor in the splenocytes derived from the tumor-free control mice. We then investigated the effects of the conditioned medium obtained from the co-culture on histamine synthesis in the other freshly prepared splenocytes derived from tumor-bearing mice. The conditioned medium obtained from the co-culture of the splenocytes derived from tumor-bearing mice and CT-26 cells was found to induce HDC in the splenocytes derived from the tumor-bearing mice, not in the control splenocytes (Figure 1b). Murine splenocytes contain two different populations in CD11b^+^ cells: one is Gr-1^high^ and the other is Gr-1^low^. Flow cytometric analysis indicated that HDC was induced in both kinds of CD11b^+^ populations (Figure 1c). Very low levels of basal HDC expression were found in the CD11b^+^Gr-1^low^ population.

### 2.2. IFN-γ Production Mainly in Spleen CD8^+^ T Cells

IFN-γ production during the co-culture period was also measured. IFN-γ production was detected only when the splenocytes derived from tumor-bearing mice were co-cultured with CT-26 cells as well as HDC (Figure 2a). Significant levels of IFN-γ were released 12 h after the onset of the co-culture and were abolished by depleting CD8^+^ T cells (Figure 2b,c).

### 2.3. Suppression of IFN-γ Production During the Co-Culture Period by Several Histamine Receptor Antagonists

Histamine receptors have been found to be involved in modulation of cytokine production from various immune cells, including T cells [14]. We then investigated the effects of histamine produced by the splenocytes on IFN-γ production during the co-culture period. Relatively high concentrations of H_1_ antagonists, pyrilamine and diphenhydramine, and H_4_ antagonists, JNJ7777120 and thioperamide, significantly suppressed IFN-γ production during the co-culture period (Figure 3a,b). An H_2_ antagonist, cimetidine, had no effects on the IFN-γ production. We then asked whether these compounds suppressed IFN-γ production by antagonizing endogenous histamine by using the splenocytes derived from tumor-bearing HDC^−/−^ mice. Unexpectedly, these compounds suppressed IFN-γ production by the HDC^−/−^ splenocytes in a similar fashion to that of the control splenocytes (Figure 3c,d). The amounts of IFN-γ production during the co-culture periods were comparable between the wild type and HDC^−/−^ splenocytes (wild type; 980 ± 300 pg/mL, HDC^−/−^; 1400 ± 680 pg/mL, *n* = 4).

### 2.4. Effects of Histamine Receptor Ligands on IFN-γ Production in the Activated Splenocytes

We then asked whether the suppressive effects of several histamine receptor antagonists should be specific for IFN-γ production induced in the immune responses against tumor expansion. Concanavalin A (Con A), which is a lectin with the potential to activate T cells, induced IFN-γ production in the wild type and HDC^−/−^ splenocytes (wild type; 830 ± 160 pg/mL, HDC^−/−^; 900 ± 240 pg/mL, *n* = 4). Pyrilamine, diphenhydramine, JNJ7777120, and thioperamide dose-dependently suppressed Con A-induced IFN-γ production in the splenocytes derived from the wild type and HDC^−/−^ mice, whereas cimetidine had no effects on it (Figure 4). Histamine exhibited partial suppressive effects on Con A-induced IFN-γ production, which reached a plateau at 1 µM. We investigated, using the antibodies raised against CD3 and CD28, whether IFN-γ production induced by T cell receptor activation should also be suppressed by these histamine receptor antagonists. As previous studies demonstrated that H_1_ receptors should be involved in augmented Th1 responses [8,15], we investigated the effects of these compounds using the splenocytes derived from Hh1r^−/−^ mice [16] in addition to Hdc^−/−^ mice. The production of IFN-γ upon T cell receptor activation was significantly suppressed by pyrilamine, diphenhydramine, JNJ7777120, and thioperamide in the splenocytes derived from BALB/c, C57Bl6, HDC^−/−^ (backcrossed to BALB/c), and Hh1r^−/−^ strains (Figure 5). The amounts of IFN-γ were comparable between the wild type and HDC^−/−^ mice (BALB/c; 5900 ± 3800 pg/mL, HDC^−/−^; 2000 ± 400 pg/mL, *n* = 4), whereas they were higher in the Hh1r^−/−^ mice than in the wild type mice (C57Bl6; 225 ± 53 pg/mL, Hh1r^−/−^; 1400 ± 45 pg/mL, *n* = 3).

### 2.5. Effects of Histamine Receptor Ligands on IL-2 Production in the Activated Splenocytes

We observed decreases in the number of aggregated splenocyte colonies in the presence of these histamine receptor antagonists, raising the possibility that they might suppress the IL-2-mediated proliferation of T cells. We then investigated the effects of these compounds on IL-2 production in the activated splenocytes. Pyrilamine, diphenhydramine, and JNJ7777120 significantly suppressed IL-2 production in the splenocytes stimulated with Con A, whereas cimetidine, thioperamide, and histamine did not affect IL-2 production (Figure 6a). IL-2 production induced upon T cell receptor activation was found to be insensitive to all these compounds except diphenhydramine (Figure 6c). The absence of HDC did not affect the potencies of these compounds (Figure 6b,d).

## 3. Discussion

Accumulating evidence suggests that MDSC may be the major source of histamine in tumor tissues, although it remains to be clarified how HDC is induced. We reproduced here the process of induction of histamine synthesis by murine myeloid cells in the tumor tissues by establishing a system in which splenocytes derived from tumor-bearing mice were co-cultured with mitomycin c-treated tumor cells (Figure 7). HDC-expressing cells were found to be CD11b^+^Gr-1^+^ cells, raising the possibility that histamine synthesis was induced in MDSC in our model. The induction of histamine synthesis was limited to the situation in which tumor immunity was established. It is likely that CT-26 cells triggered acquired immunity against themselves during the co-culture period, and thus stimulated the production of a wide variety of cytokines, including IFN-γ. As the conditioned medium may induce histamine synthesis in the splenocytes derived from tumor-bearing mice, it is likely that soluble mediators, such as cytokines, induced upon tumor immunity may act on the surface receptors on the splenocytes to induce HDC. Previous studies demonstrated that several cytokines are involved in induction of histamine synthesis. Interleukin-3 can induce histamine synthesis in murine bone marrow [3]. Injection of IL-1 into mice was found to induce histamine synthesis in various tissues, such as bone marrow, spleen, lung, and liver, as well as lipopolysaccharide (LPS) injection [17]. Several studies suggested that GM-CSF can also induce HDC in bone marrow-derived cells [6,18]. Although we could not identify the cytokines that might be responsible for induced histamine synthesis, our co-culture model will be useful for identifying the putative HDC-inducing factors.

We then tried to determine the roles of histamine produced by MDSC in our system. Among several histamine receptor antagonists, H_1_ antagonists (pyrilamine and diphenhydramine) and H_4_ antagonists (JNJ777120 and thioperamide) were found to suppress IFN-γ production, whereas an H_2_ antagonist, cimetidine, had no effects on it. Unexpectedly, these inhibitory effects were also observed in the *Hdc*^−/−^ splenocytes, indicating that the suppressive effects of these compounds are not likely associated with their abilities to compete with histamine. As no common target molecules of these compounds have been reported, it is rather difficult to identify their targets in addition to histamine receptors. These inhibitory effects were also observed when splenocytes derived from tumor-free control mice were stimulated with concanavalin A or the combination of anti-CD3 and anti-CD28 antibodies. These results suggest that shared signaling pathways leading to IFN-γ production may be suppressed by these compounds, because CD4^+^ T cells should also be involved in IFN-γ production under these conditions. Interestingly, clozapine, which can function as an H_4_ agonist [19], was found to suppress IFN-γ production in activated human peripheral blood mononuclear cells through suppression of expression of T-bet, which is the master transcription factor of Th1 differentiation [20]. A recent study reported that some benzoxazole derivatives can suppress IFN-γ production in murine CD4^+^ T cells in a similar manner to clozapine [21]. Accumulating evidence suggests that T-bet is involved in IFN-γ production both in CD4^+^ and CD8^+^ T cells, although T-bet-independent IFN-γ production was reported in CD8^+^ T cells [22,23,24]. The compounds with suppressive effects on IFN-γ production found here may also have the potential to modulate the expression levels and transcriptional activity of T-bet.

Among the compounds tested here, pyrilamine and diphenhydramine exhibited relatively strong effects. As previous studies have indicated that several H_1_ antagonists, including pyrilamine, may function as inverse agonists [25,26,27,28], we verified this possibility using *Hh1r*^−/−^ mice. Previous studies demonstrated that H_1_ receptors are involved in the promotion of Th1 responses, including IFN-γ production [8,15]. We confirmed significant levels of H_1_ and H_2_ receptor mRNA in isolated splenocytes. However, our results obtained using the *Hh1r*^−/−^ splenocytes indicated that the H_1_ receptor is unrelated to the actions of pyrilamine and diphenhydramine. Regarding the expression levels of H_4_ receptors, no or very low levels of mRNA expression were detected in the isolated splenocytes, although abundant levels of expression were confirmed in the bone marrow cells. Our findings might be inconsistent with the previous study demonstrating that murine H_4_ receptors are distributed in the hematopoietic system, including bone marrow and spleen [29]. We speculate that JNJ7777120 might suppress IFN-γ production in an H_4_ receptor-independent manner, although it should be clarified how the expression levels of H_4_ receptors in splenocytes is dynamically changed.

Suppressive effects of these histamine receptor antagonists on IFN-γ production were observed in relatively higher concentrations, indicating that a majority of the reported findings using these compounds should not be associated with suppressed IFN-γ production. This study will contribute to the development of novel therapeutic compounds targeting IFN-γ, which plays a critical role in the pathology of various diseases, such as autoimmune diseases and inflammatory bowel diseases [30,31].

## 4. Materials and Methods 

### 4.1. Materials

The following materials were commercially obtained from the sources indicated: Mitomycin C from Sigma–Aldrich (St. Louis, MO, USA); fetal bovine serum (FBS), mouse IFN-γ ELISA Ready Set Go!, mouse IL-2 ELISA Ready Set Go!, and an Alexa 546-conjugated anti-rabbit IgG antibody from Thermo Fisher Scientific (Waltham, MA, USA); Fc blocker (clone 2.4G2), an FITC-conjugated anti-CD11b antibody, an FITC-conjugated anti-Gr-1 antibody, an anti-CD3 antibody, and an anti-CD28 antibody from BD Biosciences (Franklin Lakes, NJ, USA); an anti-HDC antibody (ab37291) from Abcam (Cambridge, UK).

### 4.2. Animals 

Specific-pathogen-free 5–10-week-old male BALB/c mice and C57BL/6 mice were obtained from Japan SLC (Hamamatsu, Japan). *Hdc*^−/−^ mice [32] were backcrossed to BALB/c strain for more than 10 generations. *Hh1r*^−/−^ mice [16] were backcrossed to C57BL/6 strain for more than 10 generations. These gene-targeted male mice were used at 8–10 weeks of age. All mice were kept in a specific-pathogen-free animal facility at Okayama University. This study was approved by the Committee on Animal Experiments of Okayama University (OKU-201269; June 2012–March 2015, OKU-2015037; April 2015–March 2018, and OKU-2018178; April 2018–March 2021).

### 4.3. Murine Syngeneic Tumor Model 

Male BALB/c mice (5 weeks of age) were transplanted with a syngeneic colon tumor cell line, CT-26 (1 × 10^6^ cells/mouse, at the dorsal skin) as previously described [9]. The spleen was collected 14 days after the transplantation, minced, and filtered with nylon mesh to obtain the splenocytes.

### 4.4. Co-Culture of Splenocytes and CT-26 Cells 

Suspended splenocytes were treated with ACK buffer (150 mM NH_4_Cl containing 10 mM KHCO_3_, 1 mM EDTA) to eliminate red blood cells. CT-26 cells were treated with 1 µg/mL mitomycin C for 3 h in RPMI-1640 medium containing 10% heat-inactivated FBS, 100 U/mL penicillin, and 0.1 mg/mL streptomycin (complete RPMI medium) and then twice washed. Splenocytes suspended in the complete RPMI medium were seeded onto the monolayer of CT-26 cells.

### 4.5. Measurement of Histamine 

Splenocytes were homogenized in the cell lysis buffer (10 mM potassium phosphate, pH 6.8, containing 10 mM KCl, 1.5 mM MgCl_2_, 0.2 mM dithiothreitol, 0.01 mM pyridoxal phosphate, 1 mM EDTA, 1 mM EGTA, 0.2 mM phenylmethylsulfonyl fluoride, 0.1 mM benzamidine, 10 µg/mL aprotinin, 10 µg/mL leupeptin, 10 µg/mL E-64, 1 µg/mL pepstatin A, and 0.1% Triton X-100) and centrifuged at 10,000× *g* for 15 min at 4 °C. The resultant supernatant was subjected to the enzymatic assay of HDC. The reaction was performed in 0.1 M potassium phosphate, pH 6.8, containing 0.2 mM dithiothreitol, 0.01 mM pyridoxal phosphate, 2% polyethylene glycol #300, 0.2 mM aminoguanidine, and 0.8 mM histidine, for 4 h at 37 °C and then terminated by addition of perchloric acid (fin. 3%). Histamine was fluorometrically measured by HPLC with a cation exchange column, WCX-1 (Shimadzu, Kyoto, Japan), after derivatization with *O*-phthalaldehyde [33].

### 4.6. Flow Cytometry 

Cultured splenocytes were collected and incubated with the 2.4G2 antibody to block Fc receptors at 4 °C for 15 min. The cells were then labeled with the antibodies raised against the surface antigens, CD11b or Gr-1. For intracellular staining, the cells were fixed with PBS containing 4% formaldehyde at 4 °C for 30 min and then permeabilized with PBS containing 0.1% saponin and 1% FBS at room temperature for 10 min. The permeabilized cells were incubated with an anti-HDC antibody at 4 °C for 30 min and visualized with a phycoerythrin-conjugated anti-rabbit IgG antibody. The cells were analyzed using FACSCalibur (BD Biosciences).

### 4.7. Measurement of Cytokines 

The amounts of cytokines in the medium were measured using ELISA kits according to the manufacturer’s instructions.

### 4.8. Depletion of CD8^+^ T cells 

Depletion of CD8^+^ T cells from total splenocytes was performed using a CD8a^+^ T cell Isolation Kit II (Miltenyi Biotec, Bergisch Gladbach, Germany) according to the manufacturer’s instructions.

### 4.9. Stimulation of T Cell Receptors 

Splenic T cells were activated in the antibody-coated culture wells. Culture wells were coated with an anti-CD3 antibody (5 µg/mL) and an anti-CD28 antibody (1 µg/mL) at 4 °C overnight. In the case of the splenocytes derived from the *Hh1r*^−/−^ mice and the control C57BL/6 mice, 2 µg/mL of an anti-CD3 antibody, and 0.4 µg/mL of an anti-CD28 antibody were used for coating.

### 4.10. Measurement of Histamine Receptor mRNAs 

Total RNAs were extracted from the splenocytes using a NucleoSpin RNA Kit (TaKaRa Bio Inc., Kusatsu, Japan) and reverse transcribed using a PrimeScript™ RT Reagent Kit (TaKaRa Bio Inc., Kusatsu, Japan). First strand DNAs were subjected to quantitative PCR using a KOD SYBR qPCR Mix (TOYOBO, Osaka, Japan) or a SYBR Green PCR Master Mix (Thermo Fisher Scientific, Waltham, MA, USA) with the specific primer pairs as follows. *Hh1r*: 5′-TCA CTC CAG GCC TCA CAT GAC-3′, 5′-CAA AGT TCT CAT CCC AAG TTT CCA-3′, *Hh2r*: 5′-CAG TCC TAA GCG ACC CGG TA-3′, 5′-GGC ACT GCT GGA TGT ATC TTG A-3′, *Hh3r*: 5′-ATG ACC GAT TCC TGT CAG TCA CTC-3′, 5′-TTC CGA ACA GCC CGT CTT G-3′, *Hh4r*: 5′-TAC TGG CAT CTT GCC ACC AG-3′, 5′-ACG TGA GGG ATG TAC AGA GGA-3′, and *Gapdh*: 5′-TGT GTC CGT CGT GGA TCT GA-3′, 5′-TTG CTG TTG AAG TCG CAG GAG-3′.

### 4.11. Statistics 

Statistical significance for comparison between two groups was determined using an unpaired Student’s t-test. Statistical significance for comparisons among multiple groups was determined using a one-way ANOVA. Additional comparisons were made with a Dunnett multiple comparison test for comparison with the control groups or a Tukey–Kramer multiple comparison test for all pairs of column comparison.

## Figures and Tables

**Figure 1 ijms-19-04083-f001:**
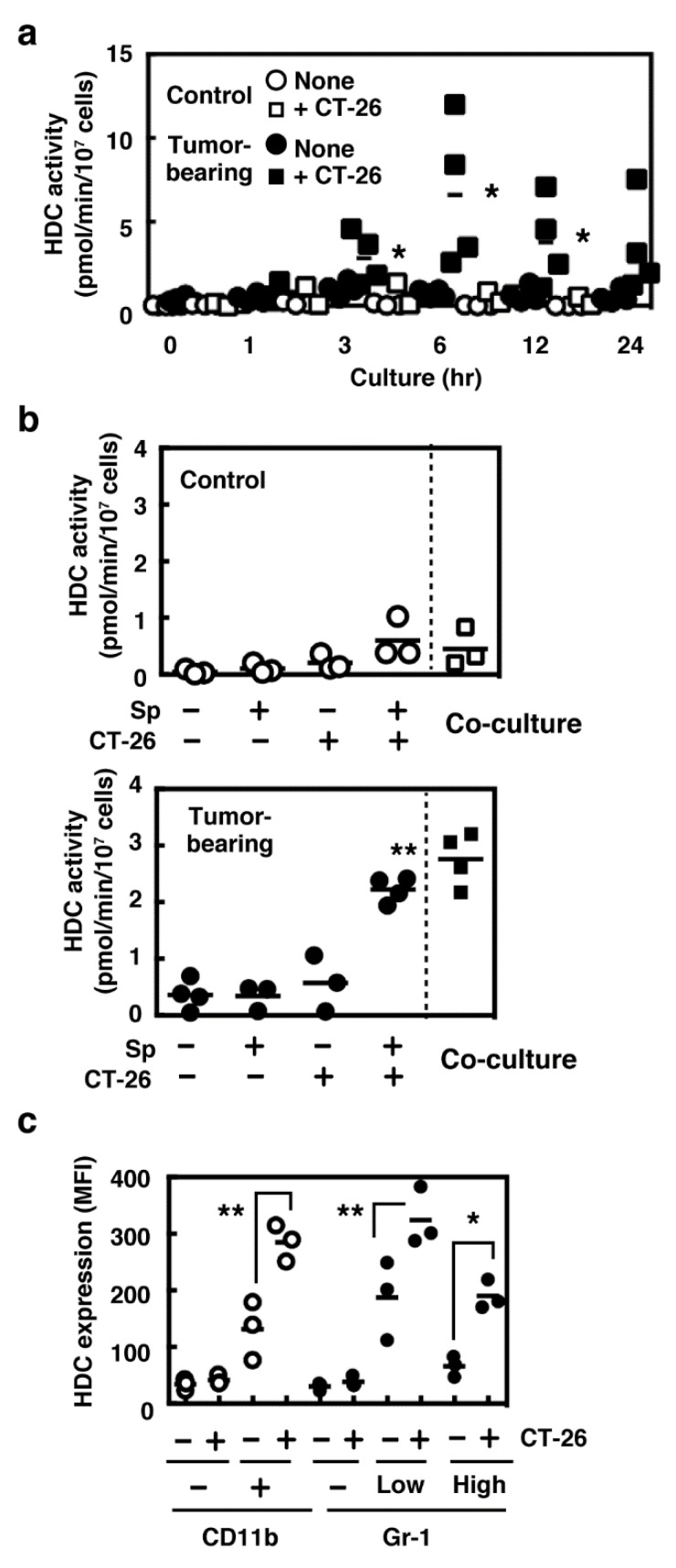
(**a**) Splenocytes were collected from the control (open symbols) and tumor-bearing mice (closed symbols) and then cultured in the absence (circles) or presence of CT-26 cells (squares) for the periods indicated. l-Histidine decarboxylase (HDC) activity was measured in the cultured splenocytes. (**b**) Cultures of splenocytes were performed under various conditions, as described above, for 6 h, and the conditioned media were then collected. Splenocytes from the control (open symbols) and tumor-bearing mice (closed symbols) were incubated in each conditioned medium for 6 h, and HDC activity was measured. HDC activity in the splenocytes co-cultured with CT-26 cells was measured as the reference. (**c**) Expression levels of HDC in the splenocytes derived from the tumor-bearing mice co-cultured with or without CT-26 cells were measured using flow cytometry with the antibodies raised against CD11b, Gr-1, and HDC. The expression of HDC was analyzed in the subpopulations of the splenocytes. Multiple comparisons were performed using one-way ANOVA with the Tukey post test. Values with * *p* < 0.05 and ** *p* < 0.01 are regarded as significant.

**Figure 2 ijms-19-04083-f002:**
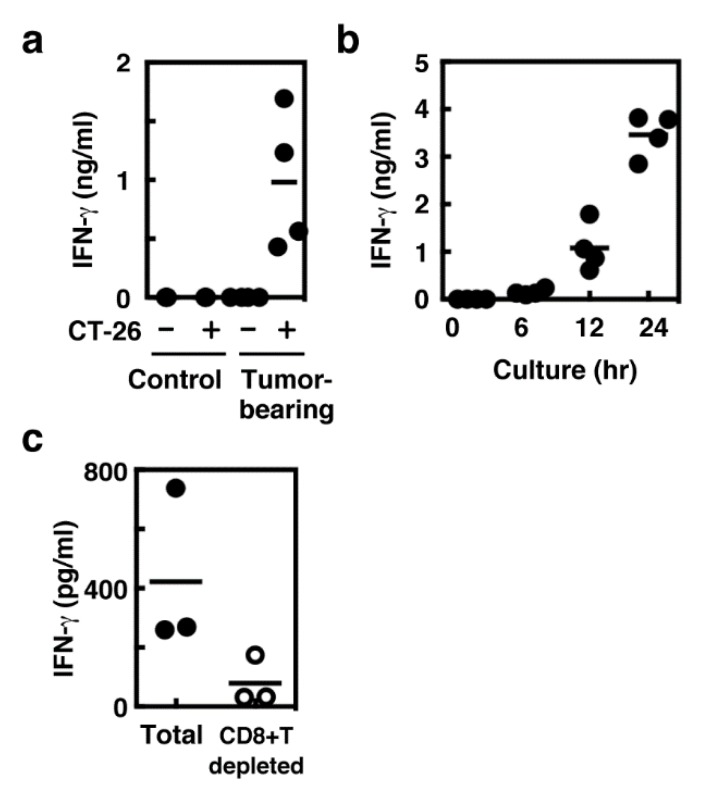
(**a**) Splenocytes were collected from the control and tumor-bearing mice and then cultured in the absence or presence of CT-26 cells for 24 h. The amounts of IFN-γ in the medium were measured. (**b**) Splenocytes were collected from the tumor-bearing mice and co-cultured with CT-26 cells for the periods indicated. The amounts of IFN-γ in the medium were measured. (**c**) Total splenocytes derived from tumor-bearing mice or those with depleted CD8^+^ T cells were co-cultured with CT-26 cells for 24 h. The amounts of IFN-γ in the medium were measured.

**Figure 3 ijms-19-04083-f003:**
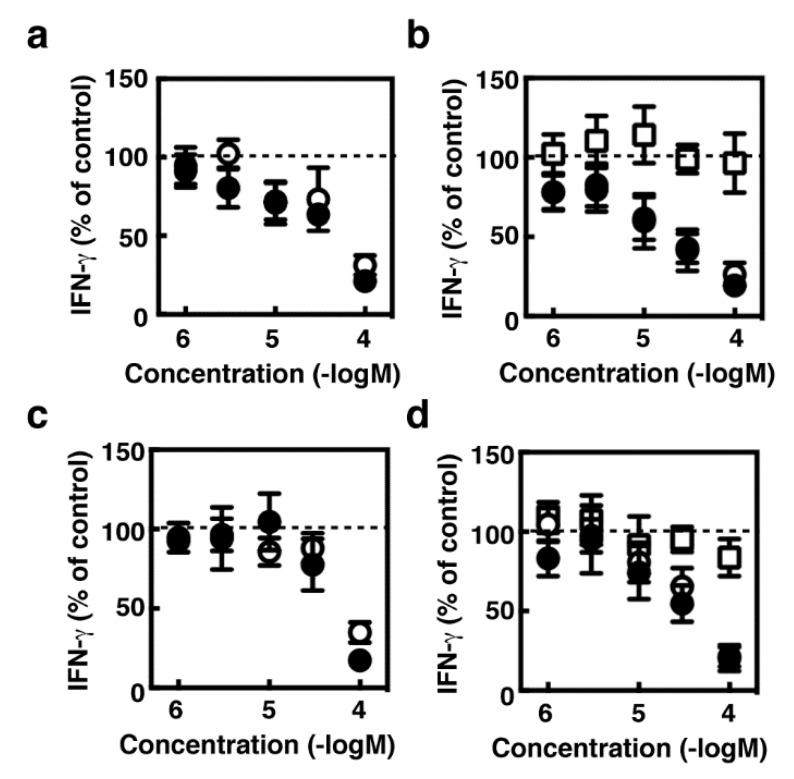
Splenocytes derived from the tumor-bearing wild type mice (**a**,**b**) and *HDC*^−/−^ mice (**c**,**d**) were co-cultured with CT-26 cells for 24 h in the presence of the indicated concentrations of pyrilamine (**a**,**c**, open circles), diphenhydramine (**a**,**c**, closed circles), cimetidine (**b**,**d**, open squares), JNJ7777120 (**b**,**d**, open circles), and thioperamide (**b**,**d**, closed circles). The amounts of IFN-γ in the medium are presented as the percentages of the control. Values are presented as the mean ± SEM (*n* = 3).

**Figure 4 ijms-19-04083-f004:**
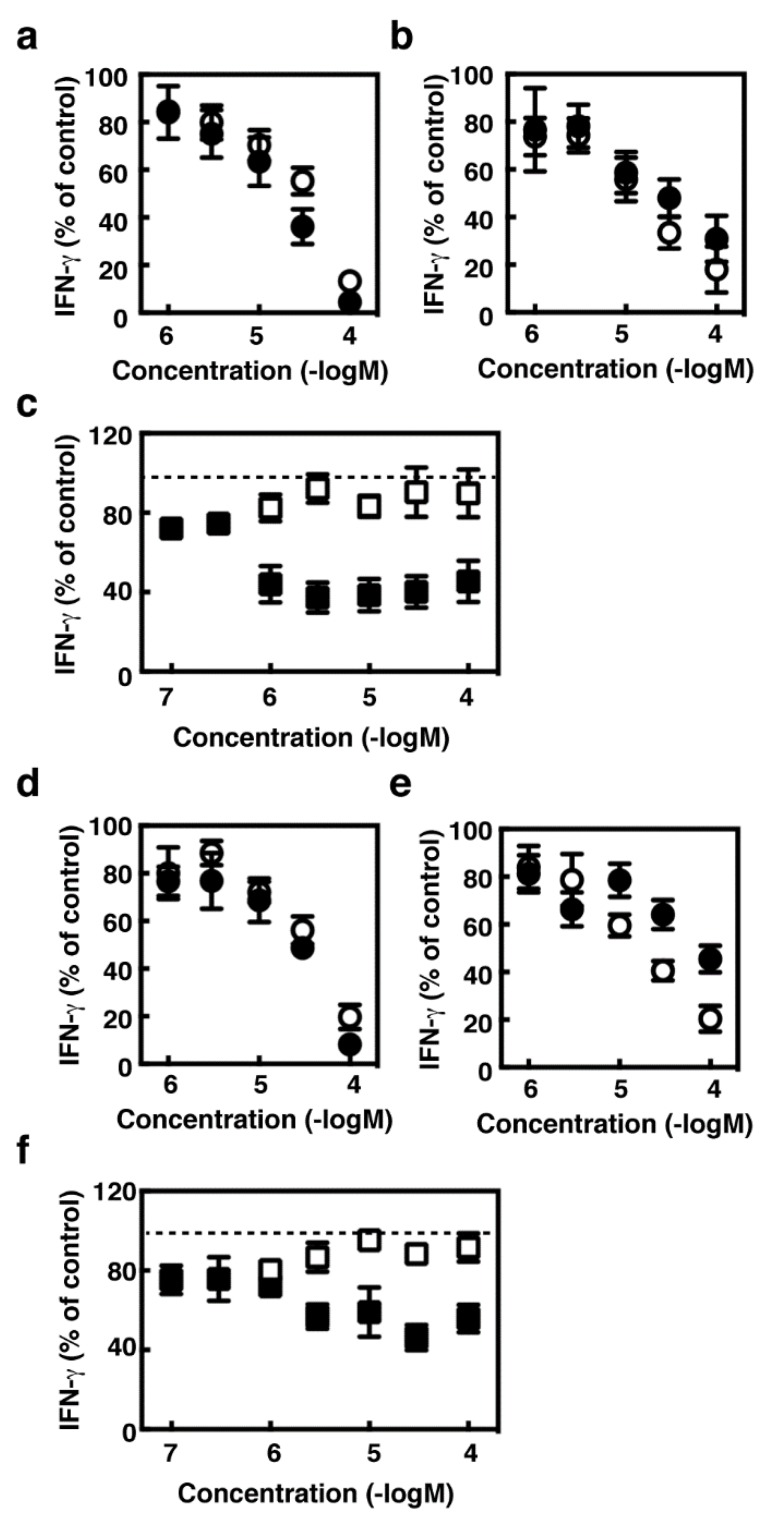
Splenocytes derived from the wild type mice (**a**–**c**) and *HDC*^−/−^ mice (**d**–**f**) were stimulated with concanavalin A (5 µg/mL) for 24 h in the presence of the indicated concentrations of pyrilamine (**a**,**d**, open circles), diphenhydramine (**a**,**d**, closed circles), cimetidine (**c**,**f**, open squares), JNJ7777120 (**b**,**e**, open circles), thioperamide (**b**,**e**, closed circles), and histamine (**c**,**f**, closed squares). The amounts of IFN-γ in the medium are presented as the percentages of the control. Values are presented as the mean ± SEM (*n* = 3).

**Figure 5 ijms-19-04083-f005:**
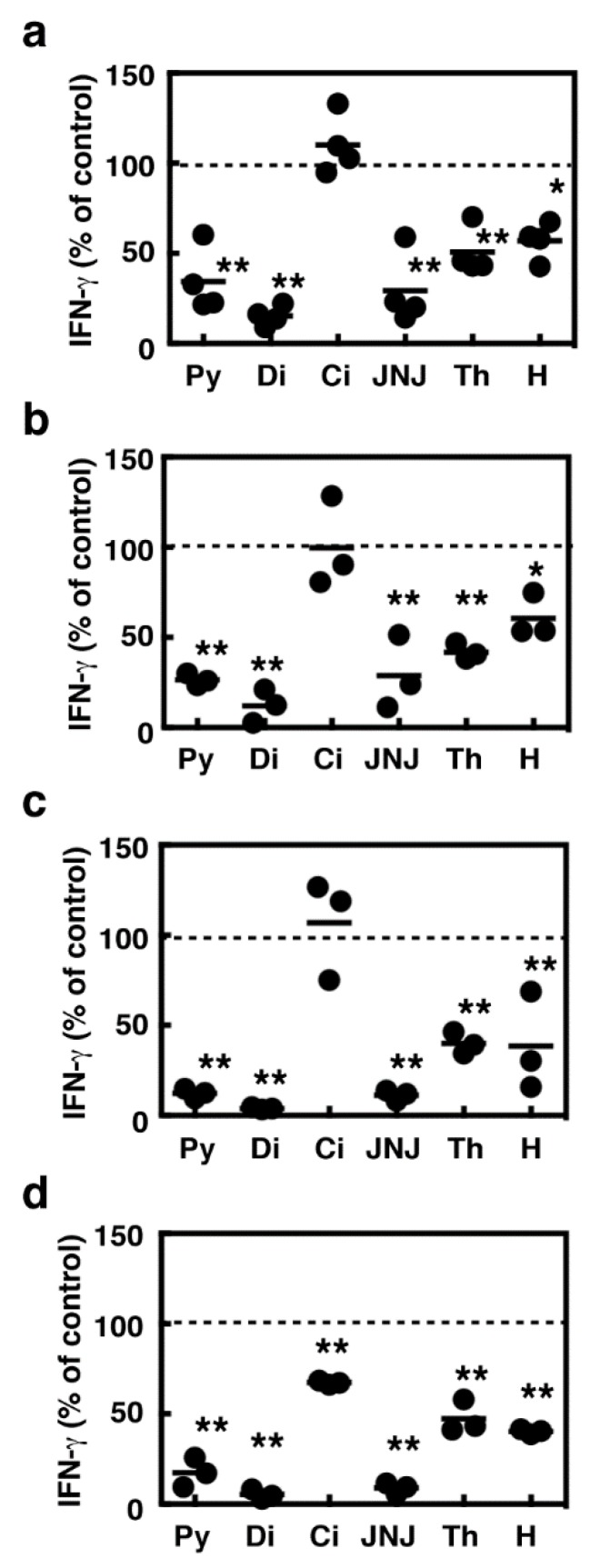
Splenocytes derived from the BALB/c mice (**a**), *HDC*^−/−^ mice (**b**), C57Bl6 mice (**c**), and *Hh1r*^−/−^ mice (**d**) were stimulated in the culture plates coated with an anti-CD3 antibody and an anti-CD28 antibody for 24 h in the presence of 100 µM of pyrilamine (Py), diphenhydramine (Di), cimetidine (Ci), JNJ7777120 (JNJ), thioperamide (Th), and histamine (H). The amounts of IFN-γ in the medium are presented as the percentages of the control. Multiple comparisons were performed using one-way ANOVA with the Tukey post test. Values with * *p* < 0.05 and ** *p* < 0.01 are regarded as significant.

**Figure 6 ijms-19-04083-f006:**
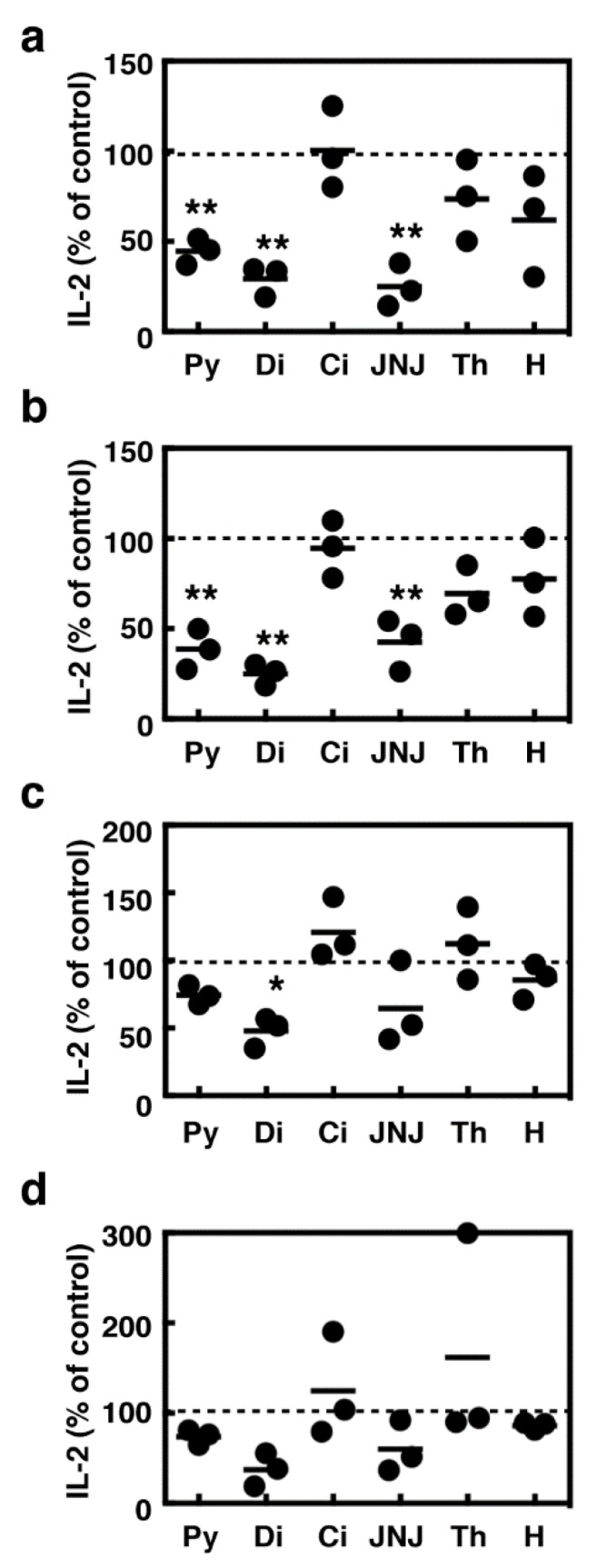
Splenocytes derived from the wild type mice (**a**,**c**) and *HDC*^−/−^ mice (**b**,**d**) were stimulated with concanavalin A (5 µg/mL, **a**,**b**) or in the culture plates coated with an anti-CD3 antibody and an anti-CD28 antibody (**c**,**d**) for 24 h in the presence of 100 µM of pyrilamine (Py), diphenhydramine (Di), cimetidine (Ci), JNJ7777120 (JNJ), thioperamide (Th), and histamine (H). The amounts of IL-2 in the medium are presented as the percentages of the control. Multiple comparisons were performed using one-way ANOVA with the Tukey post test. Values with * *p* < 0.05 and ** *p* < 0.01 are regarded as significant.

**Figure 7 ijms-19-04083-f007:**
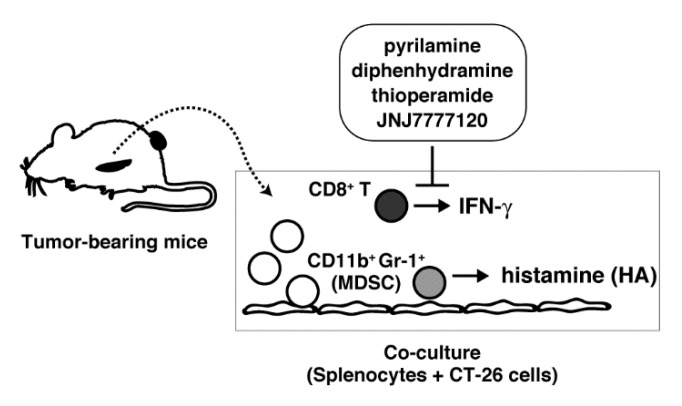
Schematic presentation of this study. MDSC; myeloid-derived suppressor cells.

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
