# Peer review of "Suppression of IFN-γ Production in Murine Splenocytes by Histamine Receptor Antagonists"

_ijms, 2018, doi:10.3390/ijms19124083_

Round 1

Reviewer 1 Report

With interest I read the paper by Kamei et al., a team of experienced researchers. I really like the idea behind the study and how it is designed. Likewise, the results are really interesting.

Still I have several comments/suggestions (no special order):

1.       It would be great if the Authors provided additional figure, scheme, a kind of graphical abstract, even more detailed that the graphical abstract required by the Journal. This figure should summarize the findings of the paper, so that everything is very clear to the reader. For example, splenocytes from mice with a syngeneic colon carcinoma co-cultured with CT-26 cells produced histamine and IFN-gamma. It should be shown in this figure what comes first, histamine or IFN-gamma, what controls what (if anything), etc.

2.       Figure 1a is barely readable as the symbols are much too small. I suggest enlarging Figure 1a to make the symbols clearer/better visible.

3.       Figure 2, the legend. It is “depleted with CD8+ T cells”. Should it not be “with depleted CD8+ T cells” (the latter seems logical based on the rest of the manuscript)? the meaning would be completely different though?

4.       Page 8, line 16. It is “limited in the situation”. Should it not be “limited to the situation” (the same comment as above)?

5.       Page 8, line 16, further. Should it not be “anti-tumor immunity” instead of “tumor immunity”?

6.       Epigenetic mechanisms play a crucial role in the regulation of T-cells and the balance between Th1 and Th2 responses, which you studied and then discussed in your manuscript, as described in https://www.ncbi.nlm.nih.gov/pubmed/28322581 and https://www.ncbi.nlm.nih.gov/pubmed/29796022. Please, refer to this topic using the two suggested papers in the Discussion.

7.       Names of the genes should always be written in italics: page 11, lines 13-17.

Author Response

We very much appreciate that the Reviewer highly evaluated our study and kindly provided us with intellectual insights that should strengthen the manuscript.

Q1. It would be great if the Authors provided additional figure, scheme, a kind of graphical abstract, even more detailed that the graphical abstract required by the Journal. This figure should summarize the findings of the paper, so that everything is very clear to the reader. For example, splenocytes from mice with a syngeneic colon carcinoma co-cultured with CT-26 cells produced histamine and IFN-gamma. It should be shown in this figure what comes first, histamine or IFN-gamma, what controls what (if anything), etc.

A1. We agree with your proposal and will add the schematic presentation. We hope that the Reviewer would take it into consideration that it is difficult to add the putative inducers of histamine and IFN-γ, because we have no convincing evidence that support the hypothesis.

Q2. Figure 1a is barely readable as the symbols are much too small. I suggest enlarging Figure 1a to make the symbols clearer/better visible.

A2. We enlarged the symbols in Figure 1 in the revised manuscript.

Q3. Figure 2, the legend. It is “depleted with CD8+ T cells”. Should it not be “with depleted CD8+ T cells” (the latter seems logical based on the rest of the manuscript)? the meaning would be completely different though?
A3. We corrected this typo according to your indication.

Q4. Page 8, line 16. It is “limited in the situation”. Should it not be “limited to the situation” (the same comment as above)?
A4. We corrected this sentence according to your suggestion.

Q5. Page 8, line 16, further. Should it not be “anti-tumor immunity” instead of “tumor immunity”?
A5. We understand what the Reviewer indicates but we suppose that the concept of tumor immunity contains the anti-tumor responses.

Q6. Epigenetic mechanisms play a crucial role in the regulation of T-cells and the balance between Th1 and Th2 responses, which you studied and then discussed in your manuscript, as described in https://www.ncbi.nlm.nih.gov/pubmed/28322581 and https://www.ncbi.nlm.nih.gov/pubmed/29796022. Please, refer to this topic using the two suggested papers in the Discussion.
A6. We agree with the Reviewer’s thought that epigenetic mechanism should play critical roles in T cell functions. We read through these suggested papers and found the significance. However, it seems difficult to avoid unnatural discussion with the citation. Because our study does not refer to T cell functions in allergic responses, we hope that the Reviewer would kindly take it into consideration.

Q7. Names of the genes should always be written in italics: page 11, lines 13-17.
A7. We corrected the expression of the gene names according to your indications.

Reviewer 2 Report

Article ‘Suppression of IFN-Gamma production in….histamine receptor antagonists’ by Kamei et al is an overall very interesting paper. The paper has all the essence of the topic which it intent to cover.

However my questions are as follows:

Why authors took CT26 cell carcinoma which is colon cancer cell. What phenotypes it associate with (such as p53- or RB- or PTEN – etc.)?

Why authors used primarily FACS for most of the experiment; HDC is cytoplasmic protein. Western should have been wisher approach. FACS of internal protein is not very reliable.

Line 36-37 is confusing  ...’ Flow cytometric analysis indicated that HDC ….in the CD11b+Gr-1low population.’ Is there are two different population of cells?

What could be the explanation of IFN-ϒ production from splenocytes? IFN-ϒ produced mainly in response to infection (bacterial or viral). Did authors also estimated other pro-inflammatory cytokines levels such as Il6, IL8, Il-1β or TNF-α etc?

Do authors believe HDC -/- mice should not produce IFN-ϒ?

Author should describe the role of Histamine in tumor progression also they should describe the potential role of histamine receptor in tumor progression in introduction or conclusion.

Author Response

We very much appreciate that the Reviewer highly evaluated the significance of this study and provided us with intellectual inputs that should strengthen the manuscript.

Q1. Why authors took CT26 cell carcinoma which is colon cancer cell. What phenotypes it associate with (such as p53- or RB- or PTEN ? etc.)?
A1. This model is one of the popular murine syngeneic tumor models. We have not paid a particular attention to the genetic background of CT-26 cells. We published two papers (reference #9 and #10) using this model and continued these studies here.

Q2. Why authors used primarily FACS for most of the experiment; HDC is cytoplasmic protein. Western should have been wisher approach. FACS of internal protein is not very reliable.
A2. We used the flow cytometric approach here to identify the cell populations that express HDC. We agree with the Reviewer that the expression of HDC should be determined by Western blot analysis. Because the excellent sell sorter is now not available for us, we could not determine the expression of HDC through immunoblot analyses of the specific splenocyte populations.

Q3. Line 36-37 is confusing  ...’ Flow cytometric analysis indicated that HDC ….in the CD11b+Gr-1low population.’ Is there are two different population of cells?
A3. Murine splenocytes contain two different CD11b+ populations; one express relatively high levels of Gr-1 and the other does low. We added the explanation in the revised manuscript.

Q4. What could be the explanation of IFN-γ production from splenocytes? IFN-γ produced mainly in response to infection (bacterial or viral). Did authors also estimated other pro-inflammatory cytokines levels such as Il6, IL8, Il-1β or TNF-α etc?
A4. We speculate that CD8+ T cells derived form tumor-bearing mice should produce IFN-γ in response to antigen presentation from CT-26 cell, although we could not demonstrate this direct interaction here. We did not measure the other pro-inflammatory cytokine levels.

Q5. Do authors believe HDC -/- mice should not produce IFN-γ?
A5. Comparable amounts of IFN-γ production were also detected in HDC-/- splenocytes and we described it in the manuscript.

Q6. Author should describe the role of Histamine in tumor progression also they should describe the potential role of histamine receptor in tumor progression in introduction or conclusion.
A6. We first tried to identify how histamine should modulate tumor immunity but found that the effects of histamine receptor antagonists were independent of histamine receptors. We therefore added in the revised manuscript that the histamine receptor subtypes involved in maturation of MDSC remains to be identified in the section of Introduction.